# The Development of iDPC-STEM and Its Application in Electron Beam Sensitive Materials

**DOI:** 10.3390/molecules27123829

**Published:** 2022-06-14

**Authors:** Hongyi Wang, Linlin Liu, Jiaxing Wang, Chen Li, Jixiang Hou, Kun Zheng

**Affiliations:** Beijing Key Laboratory of Microstructure and Properties of Solids, Faculty of Materials and Manufacturing, Beijing University of Technique, Beijing 100124, China; hongyiwang@yeah.net (H.W.); wangjiaxing26011@163.com (J.W.); chen.l_black.king@emails.bjut.edu.cn (C.L.); jixiang.hou@emails.bjut.edu.cn (J.H.)

**Keywords:** electron beam sensitive materials, electron microscopic characterization, low dose, iDPC-STEM

## Abstract

The main aspects of material research: material synthesis, material structure, and material properties, are interrelated. Acquiring atomic structure information of electron beam sensitive materials by electron microscope, such as porous zeolites, organic-inorganic hybrid perovskites, metal-organic frameworks, is an important and challenging task. The difficulties in characterization of the structures will inevitably limit the optimization of their synthesis methods and further improve their performance. The emergence of integrated differential phase contrast scanning transmission electron microscopy (iDPC-STEM), a STEM characterization technique capable of obtaining images with high signal-to-noise ratio under lower doses, has made great breakthroughs in the atomic structure characterization of these materials. This article reviews the developments and applications of iDPC-STEM in electron beam sensitive materials, and provides an outlook on its capabilities and development.

## 1. Introduction

With the development of material science, the corresponding relationship between material structures and their properties has become more precise and refined. Compared with an overall periodic crystal structure [1,2,3,4], some local structures, such as interfaces, defects, and surfaces [5,6,7], can better reflect this relationship. Since these structures are often non-periodic, real-space characterization is often required, which has been greatly facilitated by the development of electron microscopy (EM). By using spherical aberration and chromatic aberration correctors [8,9], huge improvements in image resolution have been achieved. The adoption of direct detection electron counting (DDEC) has greatly improved the detective quantum efficiency (DQE) [10,11], resulting in further improvements in imaging quality, as shown in Figure 1.

However, the electron beam with higher energy can cause some damage to the material structure while carrying the structural information of the sample [12,13]. Especially for electron-beam sensitive materials such as MOFs [14,15,16], two-dimensional materials [12], and lithium-ion batteries [17], it is extremely serious and difficult to ignore. Therefore, the use of electron microscopy to characterize the structure of such materials is both important and challenging.

Damage caused by the electron beam is inevitable, and its formation mechanism is complex. Figure 2 shows the main mechanisms of electron beam damage in STEM, including knock-on damage, radiolysis damage, electrostatic charging, and heating. During the actual characterization process, multiple mechanisms often coexist, influence, and even transform each other under certain conditions. A series of characterization methods for different types of electron beam-sensitive materials have been carried out [18,19,20,21,22,23], iDPC-STEM [24,25,26] is one of the most versatile and promising tools at present.

Figure 3 shows the number of applications of iDPC-STEM in the characterization of electron beam sensitive materials (low-dose EM techniques). It can be seen that its proportion is currently at a low level, which is greatly related to the fact that the technology is in the early stage of development and lacks the process of combining with actual characterization work. As the intersection of materials science and electron microscopy, differences in the field are also one of the reasons for its obvious advantages but limited applications.

The deepening understanding of material structure is bound to promote rapid advancement in materials and related fields. This paper will introduce the development history, imaging principle and application status of iDPC-STEM in real-space EM characterization, and summarize and prospect the application prospect of this method in atomic-scale imaging, hoping to promote the development of electron beam-sensitive materials structural characterization.

## 2. Technological Development

Phase contrast was proposed by Rose [27] in 1974. In the same year, Dekkers and De Lang [28] pointed out that the technique of obtaining images by using the relative signal difference of partition detectors could be applied to scanning transmission electron microscopy [29], which introduced the differential phase contrast (DPC) technique into electron microscopy. Waddell et al. [30] pointed out the basis for its application in electron microscopy, which could be achieved through a “first moment” detector to measure the momentum transferred to the probe and obtain the centroid of the intensity of electron beam illumination. In 1977, Rose [31] again reviewed DPC techniques using partitions in STEM and proposed execution time integrals to restore the phase. At that time, the understanding of the influence of noise and the linear relationship between the momentum transfer (centroid position) of the electron beam and the phase transfer function (phase contrast) of the sample, its physical significance, and an in-depth mathematical proof had not yet been determined. In 1978, Chapman et al. [32] began using the DPC technique for the quantitative study of magnetic structures in ferromagnetic thin films. For the next three decades, it was mainly used to study magnetic samples [33,34,35,36,37]. During this period, the theoretical research of DPC imaging was mainly based on optical microscopy, while imaging research based on electron microscopy was rarely mentioned [38].

In 2010, Shibata [39] and others once again introduced the DPC technique into electron microscopy and obtained the first convincing single-segment imaging experimental results of non-magnetic samples with atomic resolution. Since then, the DPC technique has been applied to electron microscopy. During its continuous development [40,41,42,43,44], the algorithms have been continuously optimized. The collection of local electric potential (DPC-STEM), local charge density (dDPC-STEM), and local electrostatic potential (iDPC-STEM) are all based on the COM approximate technique, and the three complement each other. In 2016, mathematical proof of the linear relationship between the COM position and the phase contrast of the sample was achieved, and the iDPC technique experimental image was provided for the first time.

## 3. Technological Principle

For non-magnetic samples, basic electrostatics stipulates that the electric field (conservative vector field) of the sample is the gradient (differential) of the electrostatic potential field (scalar field) of the sample. Electrons passing through the sample are affected by this electric field. If the sample is very thin, the electric field at the point of impact will deflect electrons in proportion to their in-plane components. This deflection can be measured by detecting the position of electrons on the far-field detector to obtain the corresponding convergent beam electron diffraction (CBED) pattern, thereby obtaining information about the position of the center of mass. The differential phase contrast (DPC) technique is used to measure the deflection of the electron (beam), i.e., the position of the center of mass [45]. When multiple zoned detectors are used, the smallest changes will be captured, and the centroid position of the direction can be calculated based on the alignment difference. This is the case with iDPC, which is the same as dDPC and DPC, which use partitioned probes to collect local signals to achieve COM position approximation.

Figure 4 shows a schematic of an iDPC-STEM. Through A-C and B-D, the COMx and COMy are obtained, then a high-quality image is obtained by a fitting calculation. This is a direct phase imaging process and is dependent on atomic number. The larger the atomic number, the stronger the scattering, and the brighter the signal. This imaging principle causes the intensity in iDPC images to be linearly related to atomic number, whereas in high-angle annular dark field (HAADF) images, the intensity is approximately proportional to the square of the atomic number; therefore, more information related to lighter elements can be obtained by using iDPC-STEM.

In addition, due to the higher electron utilization rate and the integration processes [46,47] (which remove a large part of the noise), it can achieve a high signal-to-noise (SNR) ratio under a lower dose. These levels are difficult to achieve using traditional characterization methods [48,49]: typically, the minimum electron beam dose required to obtain an image with enough SNR is 10^3^ e^−^/Å^2^, but iDPC-STEM can achieve around 10 e^−^/Å^2^.

## 4. iDPC-STEM Advantages

Enhancing imaging sensitivity is just as important as increasing the imaging resolution. Figure 5 compares the resolution of different elements using iDPC-STEM and HAADF-STEM. Both techniques are related to atomic number, and the contrast weakens upon decreasing the atomic number. However, comparing the element contrast of O, Ti, and Sr (circled in Figure 5) in the HAADF-STEM image, lighter elements are invisible or have reduced contrast, possibly below the noise level.

HAADF-STEM uses the high-angle scattered electrons to image, while iDPC-STEM uses atomic electrostatic potential [47,50] to directly image the phase. This difference in imaging principles, on the one hand, will lead to differences in image contrast: HAADF image contrast is approximately proportional to the square of the atomic number (Z^1.6–2.0^) [51], and iDPC image contrast is linearly related to the atomic number [46,52,53,54]; on the other hand, HAADF-STEM only uses high-angle scattered electrons, while iDPC-STEM, which uses almost all electrons for imaging, can obtain more signal at the same dose, thereby obtaining the characteristics that can be imaged at lower doses.

### 4.1. Advantages in Phase Contrast Imaging

Due to the use of atomic electrostatic potential to image the phase, iDPC-STEM has better sensitivity. The same goes for Electron Ptychography and 4D-STEM. Electron ptychography is a phase recovery method based on coherent diffraction imaging. The phase information of a sample is obtained through its diffraction pattern, and the phase contrast image of the sample is reconstructed.

The principle of 4D-STEM is shown in Figure 6a [55]. Based on the STEM mode, the annular electron detector is replaced by an array detector, and the entire diffraction pattern is recorded at each scanning position. Then, a camera with holes was developed to achieve simultaneous acquisition of EELS [56], as shown in Figure 6b. With the help of the 4D-STEM dataset, researchers can obtain the signal from any collection angle range, and various STEM images can be obtained through post-processing. Large data sets and complex post-processing are the main characteristics of the technique. By contrast, the several solid-state electron detectors (four in our case) employed by iDPC-STEM are two to three orders of magnitude faster than cameras used for electron ptychography and 4D STEM, especially without large data sets or cumbersome post-processing.

In this field, iDPC has obvious advantages and disadvantages: in terms of accuracy, electronic ptychography or iCOM-STEM can be used to obtain absolutely accurate the center of mass position information [46,47], while iDPC-STEM can only achieve the approximation of iCOM’s absolute accuracy by finer division (more partitions); in terms of imaging speed, iDPC has a much smaller amount of data processing: direct imaging, no huge reconstruction effort, a vast (orders of magnitude) speed advantage.

### 4.2. Advantages of Low-Dose Techniques

In the field of low-dose characterization, obtaining structural information of a series of electron-beam sensitive materials, such as zeolites [57,58,59,60,61,62,63], metal-organic frameworks [64,65,66,67,68,69,70], biomaterials [71,72,73,74,75,76], and some organic-inorganic hybrid materials is the main goal of the research. How to maintain a sufficient SNR of the image under low dose is the core issue of its development.

Figure 7 compares the capabilities of iDPC-STEM, HAADF-STEM, and K2 camera, which shows that iDPC-STEM has more comprehensive capabilities: in terms of imaging modes, STEM-based iDPC can obtain images that are easier to interpret, and the atomic number-related characteristics make it more capable of elemental resolution; the disadvantage is that the scanning time of a single image is long, which means that it is difficult to obtain real-time structures information and is susceptible to sample drift resulting in image distortion. In terms of data, iDPC technique does not have a huge dataset. In terms of equipment, iDPC technique only demands an additional four-section detector. Compared with K2 technique, which demands expensive cameras, iDPC technique has simpler equipment requirements and higher cost performance.

### 4.3. Advantages in Imaging Light Elements

Relevant structural information about light elements is often important. Nowadays, five main techniques are accessible to EM for imaging light atoms such as oxygen: low-angle annular dark field (LAADF-STEM), annular bright-field (ABF-STEM), integrated differential phase contrast (iDPC-STEM), negative Cs imaging (NCSI), and imaging STEM (ISTEM).

These techniques fall into three main categories. The first type is traditional STEM (sensitive to sample drift and scanning distortions), LAADF-STEM, ABF-STEM, and iDPC-STEM. The weak one here is LAADF-STEM, which is most optimal for mapping light atomic columns in very thin crystals only a few nanometers thick [77,78]. The second is based on a TEM imaging modality the image analysis process is complicated and often needs to be combined with simulations [79,80], and it is strongly dependent on aberrations and the sample thickness [79,81] The third one is convergent beam imaging combined with a CCD camera [82]. It is important to point out that imaging aberrations have a strong influence on the image and can even lead to contrast reversal. Probe aberrations have no influence whatsoever. (For additional details of the different techniques, please refer to the review [83]).

Their imaging principles and corresponding advantages and disadvantages are briefly summarized in Table 1. It can be seen that iDPC-STEM has certain universality and balance in the field of light element discrimination.

## 5. iDPC-STEM Application in Electron-Beam-Sensitive Materials

### 5.1. Application in Zeolites

Zeolites are widely used as acidic catalysts in the chemical and petrochemical industries due to their well-defined channels, high surface area, and tunable acidity [84,85,86,87,88,89,90,91,92,93,94]. Figure 8 shows statistics of published studies on zeolites in the past two decades. The overall trend is upward, and it displayed a faster upward trend from 2016 to 2020, which was related to the structural characterization of zeolites by iDPC-STEM. Although STEM images are easier to interpret than HR-TEM images, zeolite with radiolysis damage and electrostatic charging as the main damage mechanisms is more vulnerable to damage due to the instantaneous high intensity electron beam used in STEM. Therefore, HRTEM has been the main EM characterization method for many years. However, the ultra-low-dose properties of iDPC-STEM make the acquisition of high-resolution STEM images a reality.

ZSM-5 is a medium pore zeolite containing ten-membered rings. The framework is composed of two intersecting channel systems. The short axis is 5.1–5.2 Å; the other is a “Z”-shaped transverse channel with a nearly circular cross-section and a pore diameter of 5.4 ± 0.2 Å. Its unique microporous structure makes ZSM-5 have excellent shape selectivity and is a commonly used catalyst in the chemical field. ZSM-5 can be used in a variety of aromatization reactions, such as the methanol-to-aromatics process (MTA). The methanol-to-aromatics (MTA) process is important for converting coal/natural gas to chemicals [95]. Due to its poor electron beam stability, it is difficult to analyze its microstructure by transmission electron microscopy [96,97,98]. In 2016, its deactivation mechanism during the MTA reaction was explored by Prof. Wei’s group by low-dose iDPC-STEM [92]. Figure 9a is a schematic diagram of the catalytic process in MTA. Figure 9b,c is the low-dose HRTEM and iDPC-STEM images. It can be seen that the use of iDPC-STEM technique can achieve more precise characterization of the skeleton structure and explore the inside of the pore. This work has a great breakthrough in the EM characterization of ZSM-5. The results show that hydrothermal deactivation was mainly caused by the loss of frame-work acid sites and the blockage of channels by extra-framework alumina, provides theoretical guidance for further optimization of MTA process.

Due to the electron beam sensitivity of ZSM-5, it is difficult to distinguish whether the structural loss is caused by the reaction or the EM characterization. Figure 10a is an iDPC-STEM image of ZSM-5 along the <105> direction. Figure 10b is a magnified image in which the block units (Si-O islands) are bridged by only O atoms, and the O bridges can be clearly distinguished both in the image and the atomic model (in red and blue, respectively). The resolution was further confirmed by the strength profile analysis in Figure 10d. Figure 10e shows the atomically flat (010) surface with a half-channel termination structure. An enlarged atomic-scale surface structure is shown in the illustration, where even the Si-O dangling bonds on the surface can be directly observed. The results show that under the characterization condition, the atoms and dangling bonds on the surface did not undergo significant damage, which proved that iDPC-STEM can achieve the perfect structure characterization of such materials.

The iDPC-STEM technique can carry out the real structural characterization of the ZSM-5 structure, on this basis, the more in-depth study of the structure has been carried out. Figure 11a shows three particles of ZSM-5 that were closely connected by van der Waals interactions. Comparing the iDPC-STEM image at different scales (Figure 11b–e), the three particles showed completely consistent crystal orientation. The corresponding FFT patterns (Figure 11f,g) also exhibited a single-crystal-like nature with discrete diffraction spots. Such strict assembly and interfacial matching mean that the system must contain definite directional interaction, such as hydrogen bonds and hydroxyl groups. These will increase the transfer distance of guest molecules along the b-axis and ultimately affect the catalytic performance [94].

Figure 12a shows an iDPC-STEM image and its partial magnification of a Mo-ZSM-5 sample with a Si: Al ratio of 40 [99]. Previous studies have suggested a one-to-one correspondence between the Mo cluster and Al sites in zeolite frameworks when the Si/Al and Al/Mo ratios are proper [100,101] However, the correspondence was dubious due to the difficulty of real-space imaging and a lack of direct evidence. Now it is performed with iDPC-STEM. Figure 12b show zoomed-in areas (1, 2, and 3 of (a)): empty channel (Figure 12(bI)) and a channel containing a Mo cluster bound at the T_8_ site (Figure 12(bII)) and T_1_ site. (Figure 12(bIII)). According to simple statistics derived from 100 channels, Al most preferentially occupied the T_1_ site (40%), followed by T_2_ and T_5_ sites, and disfavored the T_3_ and T_6_ sites. The clear determination of the location of the acidic site (as the site of the catalytic reaction) is of great significance for the control of the catalytic behavior. In this work, iDPC-STEM was used to image the adsorbate in the pore to achieve the localization of the acidic site (aluminum) in the framework.

As shown in Figure 13 [102], a para-xylene (PX) molecule was used as a rotating pointer to detect the host-guest van der Waals interactions in the straight channel of the MFI-type zeolite framework. For a long time, single-molecule imaging was challenging but highly beneficial for investigating intermolecular interactions at the molecular level [103,104,105,106,107,108]. Now, with the help of iDPC-STEM, this has become a reality. In the iDPC-STEM image in Figure 13a, spindle-shaped spots can be clearly observed in some channels. The intensity profiles in Figure 13c also confirmed this. The spindle-shaped spots and the orientations of the PX molecules can be seen more clearly in the magnified image in Figure 13b. The agreement between the experimental and simulated results indicates that the C6-rings in the PX molecules have specific orientations, as expected in this van der Waals compass. This work achieved the direct imaging of a single molecule by iDPC-STEM to explore host-guest interactions in a range of organic-inorganic systems. This provides a new approach to studying the different behaviors of individual molecules.

Silicoaluminophosphate (SAPO) molecular sieves with small pores, such as SAPO-34 and SAPO-18, play an important catalytic role in methanol-to-olefin (MTO) reactions. Among them, molecular sieves with SAPO-18/SAPO-34 eutectic structure often show better performance than single molecular sieves in catalytic reactions due to the presence of both AEI and CHA structural units. However, due to the difficulty of electron microscopy characterization, SAPO-18 The /SAPO-34 eutectic structure is less studied. In Figure 14a–d, a local enlargement of an iDPC-STEM image of SAPO-18 (AEI) and SAPO-34 (CHA) from different orientations reveals the projection of the different bond lengths of Al-O and P-O bonds in different directions combined with their intensity profiles. This provides new possibilities for studying the symbiosis and accumulation of zeolite skeleton from the perspective of atoms and bonds. iDPC-STEM can also be used to analyze the surface atomic arrangement in thin regions. The red and blue arrows in Figure 14e indicate stacked AA or AB in SAPO-34/18 intergrowth catalyst surface. By comparing the top and bottom (Figure 14f,g), the stacking sequence was different at the top and bottom, which revealed the crystal growth trend [109].

In the field of zeolite, iDPC-STEM has achieved the characterization of fine structure. While providing real-space evidence that was difficult to obtain in the past, it has carried out a more in-depth analysis of the relationship between structure and catalyt-ic performance, which has a relatively complete and systematic study and has great development potential.

### 5.2. Application in MOFs

Metal-organic frameworks (MOFs) are typical porous materials constructed by orderly splicing organic linkers between metal nodes. Due to their designable topology, porosity, and functionality, they exhibit excellent performance for gas storage and separation, catalysis, drug delivery, and biomedical analysis [110,111]. Figure 15 shows a summary of the studies on MOFs in the past two decades, which shows a rapid upward trend.

MIL-101, the specific surface area is mostly above 4000 m^2^/g, and there is an effective inner diameter of about 2.9 nm and a window diameter of about 2.9 nm. Pentagonal holes with a diameter of about 1.2 nm and hexagonal holes with an effective inner diameter of about 3.4 nm and a window diameter of about 1.4 nm. MIL-101 is commonly used to support various metal atoms or active particles in catalysis research. Therefore, if the local structure of MIL-101 (the coordination relationship of nodes and linkers, etc.) can be imaged directly at the atomic scale, it would help understand the relationship between catalyst structure and performance [110,111]. Figure 16 shows the development of MIL-101 characterization techniques. In 2005, attempts were made to obtain low-dose HRTEM images with a certain signal-to-noise ratio. Only the main channels/cages could be distinguished [112]. In 2016, high-resolution HAADF-STEM images of heavy elements made it possible to identify the distribution of metal nodes and heavily-doped elements in MOFs [113]. In 2018, the camera was upgraded again (direct detection electron counting, DDEC). A higher detective quantum efficiency (DQE) enabled HRTEM to maintain an adequate signal-to-noise ratio at a sufficiently low electron beam dose [114]. In 2019, iDPC-STEM emerged for MOF characterization [115]. It can be seen from Figure. 16d that iDPC-STEM has a higher image contrast to the cage structure

In recent years, with the gradual popularization of iDPC-STEM, it has played an increasingly important role in the characterization of the surface, interface, and defects of MIL-101 and other non-periodic local structures [114]. Figure 17 shows the various surface structures of MIL-101 in iDPC-STEM images. This further proves the ability of iDPC-STEM in resolving cage structures.

Figure 18 shows a study on the structural evolution characteristics of MIL-101 under electron beam radiation [116]. By collecting iDPC images from different cumulative electron beam doses, the evolution of the sample was analyzed. The results showed that the sample tended to shrink after irradiation. iDPC images with a resolution of 4.7 Å and good contrast for light elements can be used to quantitatively observe and analyze the local structural evolution of MIL-101 under electron beam irradiation. This work utilizes the high contrast of iDPC image for light elements to quantitatively observe and analyze the local structural evolution of MIL-101 under -electron beam irradiation.

In 2020, a study [117] (Figure 19) proposed a new strategy called “molecular compartments”, in which iDPC was used to identify the exact location of cage structures in TiO_2_-MIL-101 composites. The relative cage positions were obtained by HAADF-STEM (Figure 19a–c). The exact positions of TiO_2_ in MIl-101 cages were characterized by iDPC-STEM (Figure 19d–f), with a resolution of 3.2 Å. It is very important to accurately characterize the position of TiO_2_ in real space, and the catalytic effects of the compartments formed by TiO_2_ in different cage structures are also different. This work utilizes iDPC-STEM to achieve atomic structure imaging of different cage structures.

In the field of MOFs, more refined structural characterization has been achieved by iDPC-STEM, and related research on boundary, load and other types has also been carried out. Compared with the low-dose HRTEM images obtained with the k2 camera, it can be seen that iDPC-STEM has a huge advantage in image contrast, which enables it to identify cage structures more accurately.

### 5.3. Application in Perovskites

For the perovskite ABO_3_, the octahedral oxygen atoms surround a central B atom forming BO_6_, where the oxygen atoms affect various degrees of freedom (spin, orbit, charge) of the B atom and can give rise to new physical properties. To understand the physical properties more fully, it is necessary to describe the atomic space occupation, including the atomic-scale interface. Figure 20 summarizes studies on perovskites reported in the past two decades, which show the same rapid upward trend as the above materials.

Figure 21 shows a direct iDPC-STEM image of oxygen and cationic columns and follows their evolution with temperature in epitaxial Hf_0.5_Zr_0.5_O_2_(HZO)/La_0.67_Sr_0.33_MnO_3_ (LSMO and bottom electrode) heterostructures [118]. Figure 22 shows the oxygen positions in BaTiO_3_ obtained by iDPC-STEM, which are often difficult to identify [119]. The lower-left insets are magnified images selected from the white dotted box areas and show a schematic of BaTiO_3_.

Solar cells based on organic-inorganic hybrid perovskite materials (OIHPs) have developed rapidly due to their high photoelectric conversion efficiency. Over the past decade, their efficiencies have soared from 3.8% to 25.2%, almost matching that of monocrystalline silicon solar cells. However, their commercial applications are limited by the structural instability of hybrid perovskite materials, high temperature, oxygen, humid environment, light, and other factors that accelerate material decomposition. Therefore, it is necessary to deepen the understanding of degradation mechanisms to guide device design and material synthesis. As an electron-beam-sensitive material, studies have shown that the electron beam damage mechanism of OIHPs is irradiation decomposition, which is mainly affected by the dose. Therefore, iDPC-STEM will be of great help. However, most studies on its electron beam damage have focused on determining periodic structures by diffraction (Table 2). The application of iDPC-STEM in this field is relatively scarce, giving it large amounts of development potential

## 6. Summary and Outlook

Here, we have briefly introduced iDPC-STEM, an ideal tool for electron-beam-sensitive material characterization, including its development, imaging principles, and main applications. At present, iDPC-STEM has excellent characterization advantages for electron-beam-sensitive materials, but its further applications face some limitations.

Since its images rely on interactions between a focused electron beam and the electric field of a sample, it is demanding on both the beam and the detector, as with STEM. It is necessary to cooperate with the spherical aberration corrector to control the size of the beam to achieve a high resolution. For the detector, more continuous refinement partitions will bring it closer to iCOM imaging capabilities.

Due to its high contrast towards light elements, it is sensitive to sample surface contaminants, especially organic matter, and amorphous carbon. Therefore, there are requirements for sample preparation, storage, and electron microscope cleanliness.

Since its imaging process is based on weak phase approximation, for thin materials, high-quality images can be obtained; however, as the thickness of a sample increases, its contrast decreases. For samples of unknown thickness, such contrast changes make in-depth analysis difficult.

Due to its strong signal receiving efficiency, low-frequency noise may be amplified during integration. Therefore, to eliminate amplified noise, iDPC-STEM often chooses adaptive filtering techniques. The development of a suitable filtering technique is important for iDPC-STEM.

At present, the technique is still in its early development stages, and there are relatively few auxiliary tools (such as simulations), which need to be perfected.

In a word, although iDPC-STEM has excellent characterization advantages for electron beam sensitive materials, it has many limitations, especially for materials scientists with a weak background in electron microscopy, which greatly reduce its practical applications. The development direction of iDPC-STEM must be to constantly eliminate the above limitations and play a more important role.

## Figures and Tables

**Figure 1 molecules-27-03829-f001:**
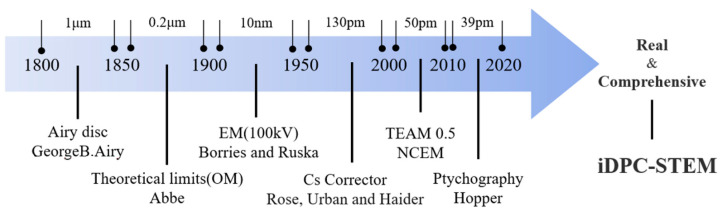
The history of microscope development: more comprehensive and more realistic structural information is the current mainstream development direction.

**Figure 2 molecules-27-03829-f002:**
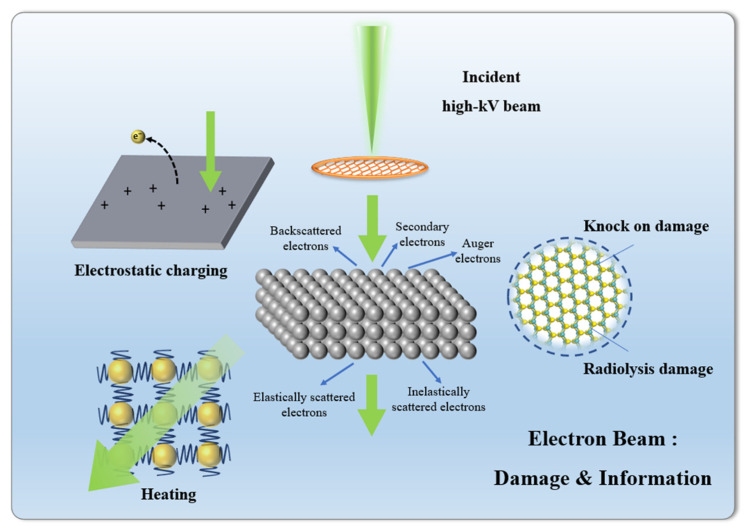
High-energy electron beams will inevitably introduce damage while reducing structural information: knock on damage, radiolysis damage, electrostatic charging, and heating.

**Figure 3 molecules-27-03829-f003:**
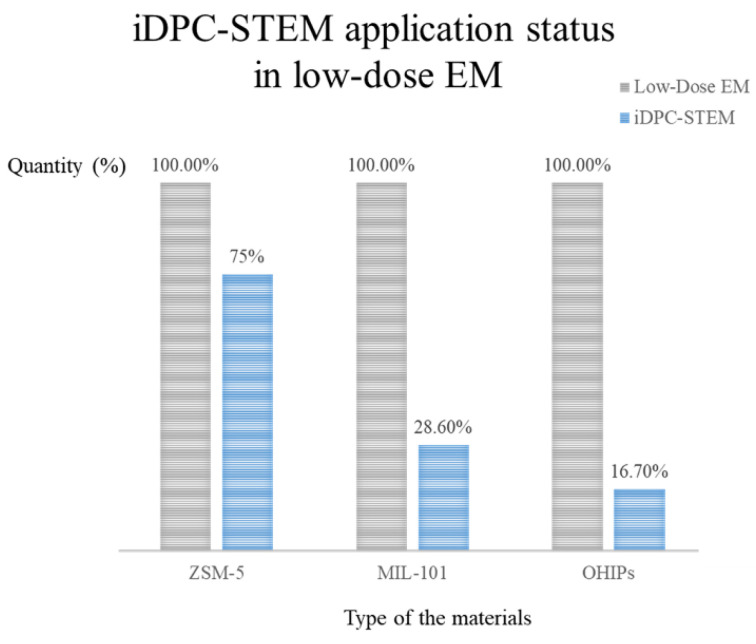
Application status of iDPC-STEM in electron beam sensitive materials characterization.

**Figure 4 molecules-27-03829-f004:**
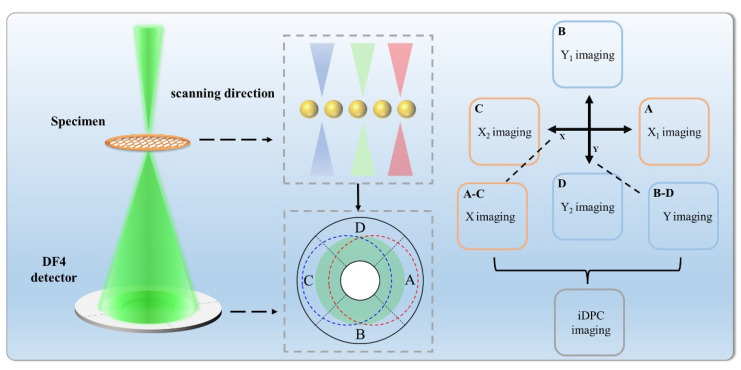
The schematic diagram of technical principle: partition probe, electrostatic potential imaging, and integral.

**Figure 5 molecules-27-03829-f005:**
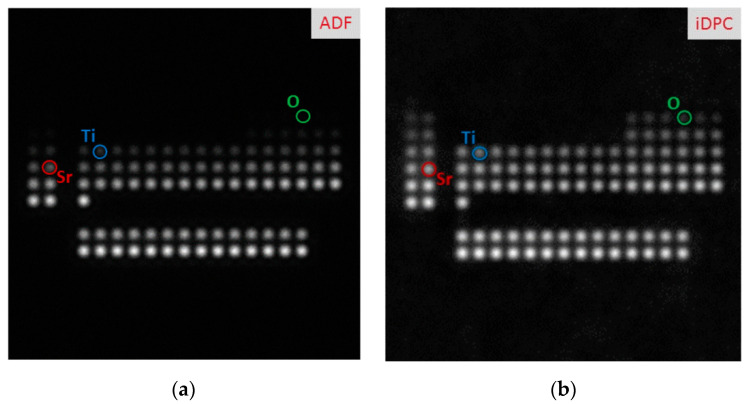
The comparison of single-atom contrast in the range Z = 1–103 obtained by simulations: HAADF-STEM image (**a**) and iDPC-STEM image (**b**) Reprinted/adapted with permission from Ref. [24]. Copyright 2016, Cambridge University Press.

**Figure 6 molecules-27-03829-f006:**
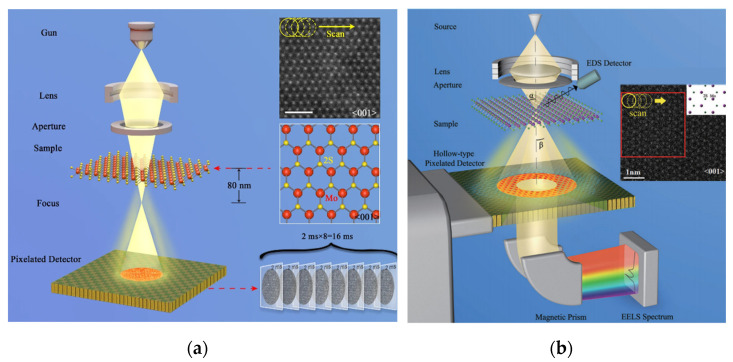
(**a**) Schematic diagram of the optical path of the 4D-STEM. Reprinted/adapted with permission from Ref. [55]. Copyright 2019, Springer Nature. (**b**) Based on the STEM mode, the ring-shaped electron detector is replaced by an arrayed detector, and the entire diffraction pattern is recorded at each scanning position Reprinted/adapted with permission from Ref. [56]. Copyright 2018, American Physical Society.

**Figure 7 molecules-27-03829-f007:**
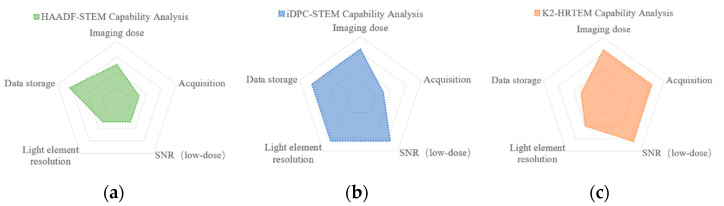
The capability comparison of HAADF-STEM (**a**), iDPC-STEM (**b**) and K2 camera (**c**).

**Figure 8 molecules-27-03829-f008:**
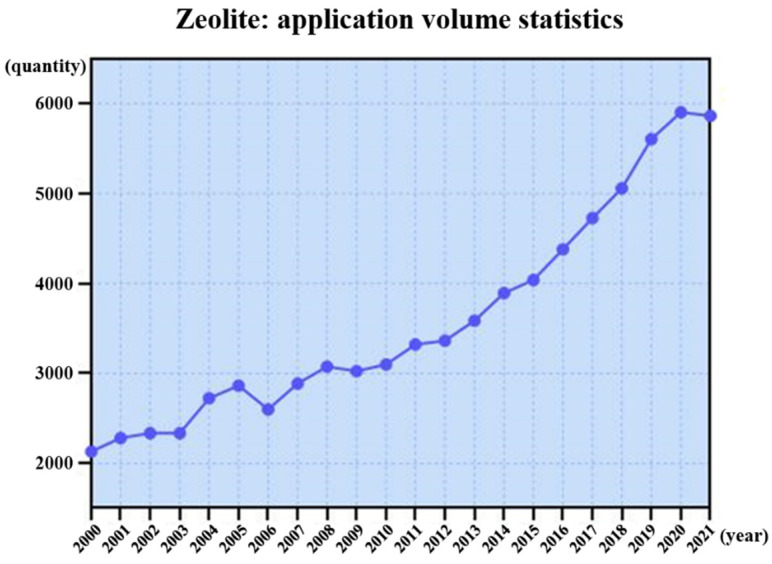
The research quantity curve of zeolite in the past two decades.

**Figure 9 molecules-27-03829-f009:**
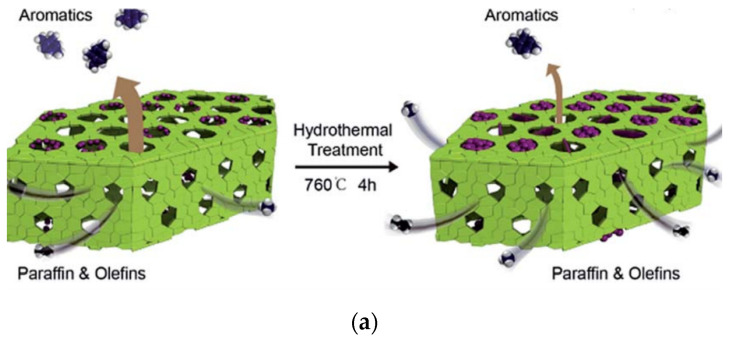
(**a**) MTA schematic; (**b**) HAADF-STEM of the synthesized ZSM-5; (**c**) iDPC-STEM Reprinted/adapted with permission from Ref. [92]. Copyright 2011, Royal Society of Chemistry.

**Figure 10 molecules-27-03829-f010:**
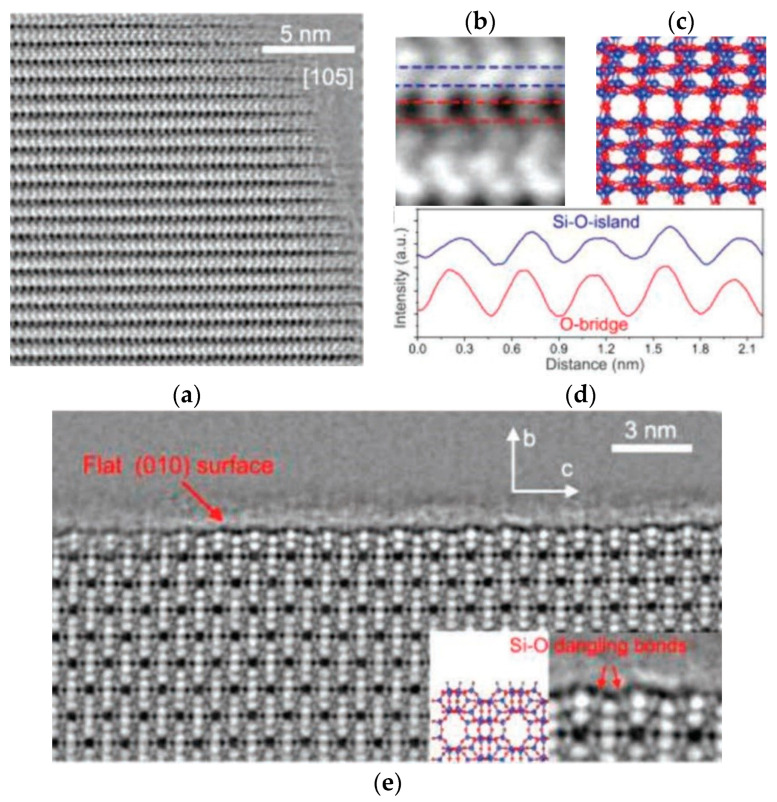
(**a**) The iDPC-STEM image of ZSM-5 from the <105> direction; (**b**–**d**) detailed analysis from (**a**); (**e**) the atomically flat (010) surface with clear Si-O dangling bonds. Reprinted/adapted with permission from Ref. [94]. Copyright 2019, Advanced Materials.

**Figure 11 molecules-27-03829-f011:**
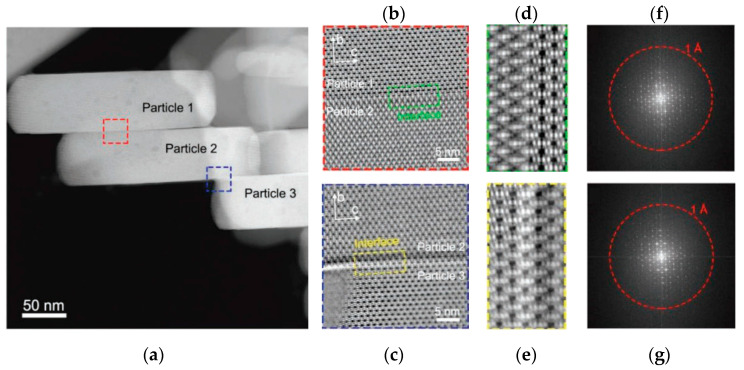
(**a**) The HAADF-STEM image of assembled ZSM-5 particles; (**b**,**c**) The iDPC-STEM images of the (010) interfaces in the areas marked in (**a**); (**d**,**e**) The zoom-in interface areas of iDPC-STEM images marked in (**b**,**c**); (**f**,**g**) The FFT patterns corresponding to (**b**) and (**c**), respectively. Reprinted/adapted with permission from Ref. [94]. Copyright 2019, Advanced Materials.

**Figure 12 molecules-27-03829-f012:**
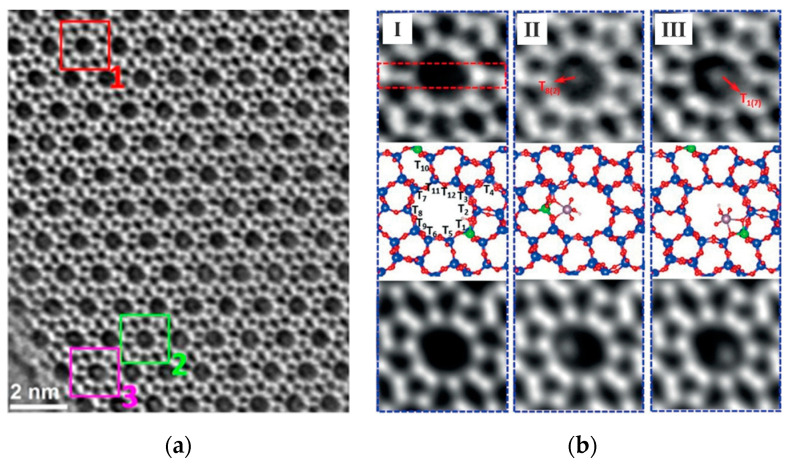
(**a**) iDPC-STEM image of Mo/ZSM-5; (**b**) Zoomed-in areas 1 (I), 2 (II), and 3 (III) of (**a**) with an atomic model. Reprinted/adapted with permission from Ref. [99]. Copyright 2019, Angewandte Chemie.

**Figure 13 molecules-27-03829-f013:**
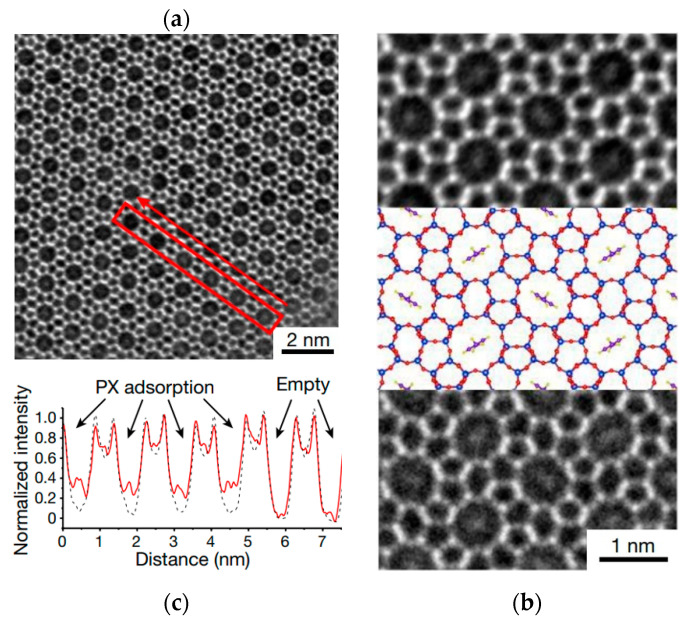
(**a**,**b**) iDPC-STEM image (**b**, **top**), the structural model (**b**, **middle**) and the simulated image (**b**, **bottom**); (**c**) shows the corresponding intensity profile acquired from the red-framed region in (**a**).Reprinted/adapted with permission from Ref. [102]. Copyright 2021, Springer Nature.

**Figure 14 molecules-27-03829-f014:**
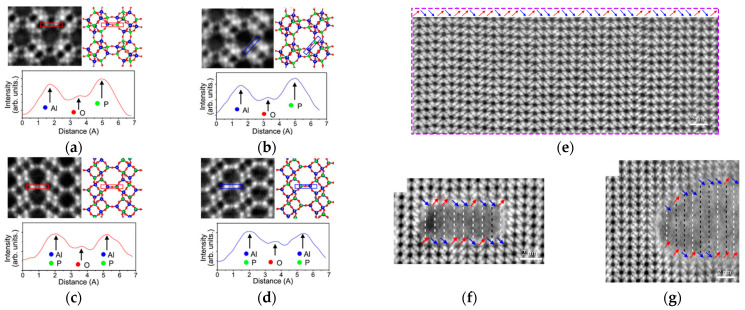
(**a**–**d**) Magnified iDPC-STEM images with red box and intensity profiles (for red box) of SAPO-18 (**a**,**c**) and SAPO-34 (**b**,**d**)with blue box; (**e**) iDPC-STEM image of highly mixed SAPO-34 and SAPO-18 lattices inside; (**f**,**g**) iDPC-STEM images show the stacking sequences. Reprinted/adapted with permission from Ref. [109]. Copyright 2021, Applied Physics Letters.

**Figure 15 molecules-27-03829-f015:**
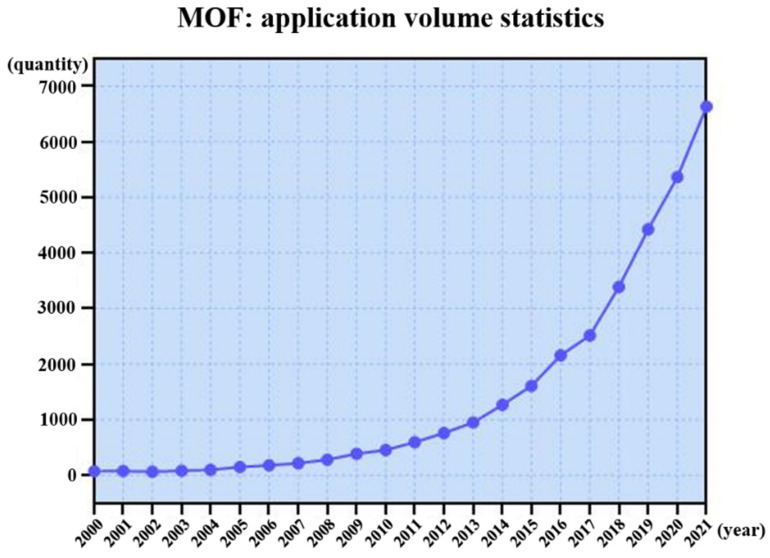
The research quantity curve of MOF in the past two decades.

**Figure 16 molecules-27-03829-f016:**
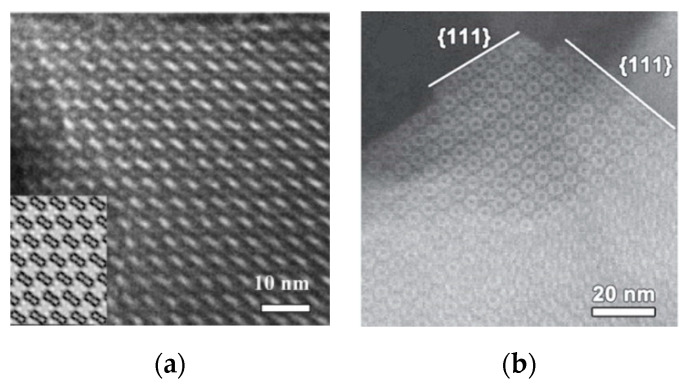
A series of electron microscopy images of MIL-101 acquired in different years: (**a**) HRTEM image in 2005. Reprinted/adapted with permission from Ref. [112]. Copyright 2005, American Chemical Society. (**b**) ADF-STEM image in 2016. Reprinted/adapted with permission from Ref. [113]. Copyright 2016, John Wiley and Sons. (**c**) low-dose HRTEM image (with DDEC) in 2019.(FFT for the white box) Reprinted/adapted with permission from Ref. [114]. Copyright 2019, American Chemical Society. (**d**) iDPC-STEM image in 2020. Reprinted/adapted with permission from Ref. [115]. Copyright 2020, Springer Nature.

**Figure 17 molecules-27-03829-f017:**
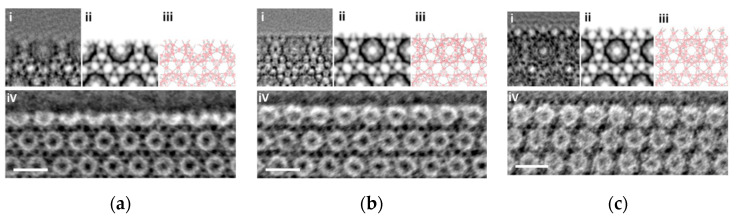
A series of electron microscopy images of the different surfaces in MIL-101: (**a**) The boundary includes most of the cage structure; (**b**) The boundary includes the whole of the cage structure; (**c**) The boundary includes few of the cage structure. And (**i**) HRETM; (**ii**) simulated image; (**iii**) structure model; (**iv**) iDPC-STEM. Reprinted/adapted with permission from Ref. [114]. Copyright 2019, American Chemical Society.

**Figure 18 molecules-27-03829-f018:**
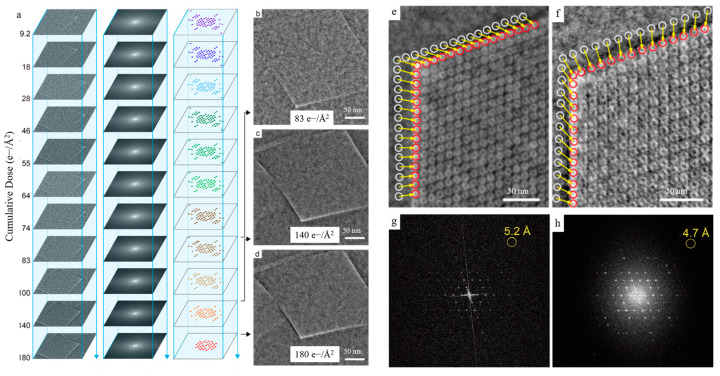
(**a**) a series of iDPC-STEM images with the increased cumulative electron beam doses; (**b**–**d**) iDPC-STEM under different dose; (**e**,**f**) iDPC-STEM images before and after irradiation; (**g**,**h**) the corresponding FFT. Reprinted/adapted with permission from Ref. [116]. Copyright 2020, American Chemical Society.

**Figure 19 molecules-27-03829-f019:**
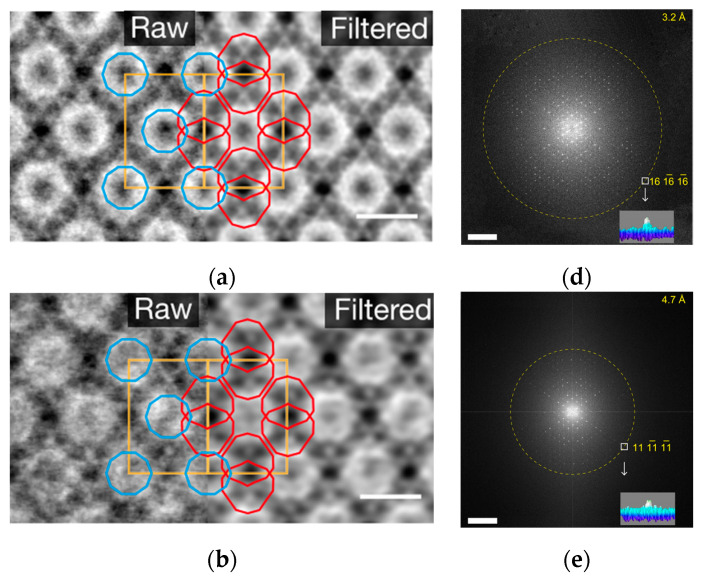
(**a**–**c**) A series of iDPC-STEM images with increased loading (TiO_2_); (**d**–**f**) Increased loading (TiO_2_) corresponding FFT. Red and blue outlines are overlaid on the images to highlight the positions of TiO_2_ units in separate types of mesopores. Scale bars: 5 nm. Reprinted/adapted with permission from Ref. [117]. Copyright 2020, Springer Nature.

**Figure 20 molecules-27-03829-f020:**
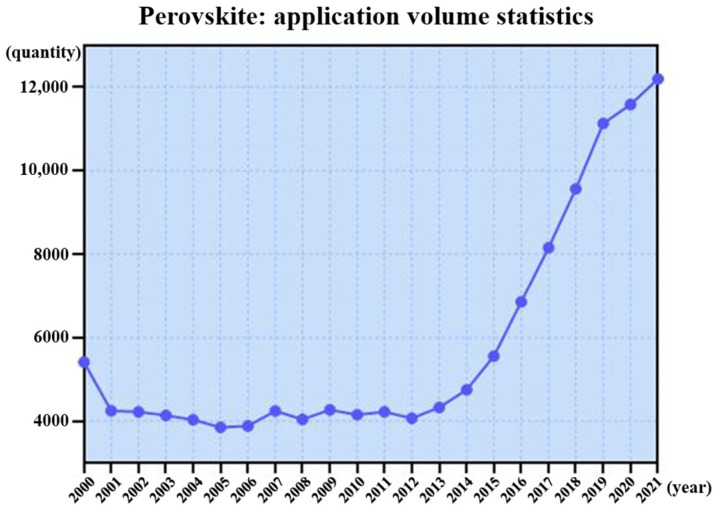
The research quantity of perovskites in the past two decades.

**Figure 21 molecules-27-03829-f021:**
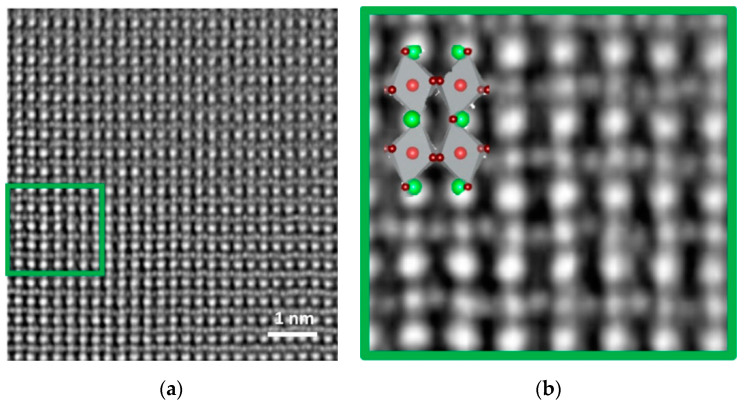
(**a**) iDPC-STEM image of the LSMO layer in the virgin state; (**b**) is the enlarged image of the green box in (**a**). Reprinted/adapted with permission from Ref. [118]. Copyright 2021, AIP Publishing.

**Figure 22 molecules-27-03829-f022:**
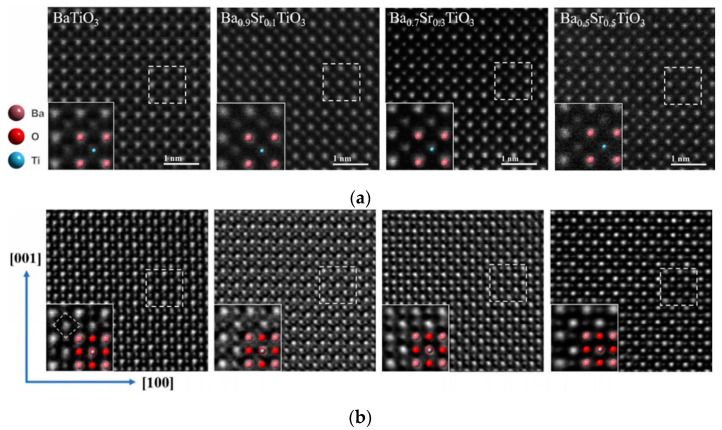
(**a**) HAADFSTEM images of selected areas; (**b**) corresponding iDPC-STEM images. The white box is to highlight and magnify the structure. Reprinted/adapted with permission from Ref. [119]. Copyright 2021, Springer Nature.

**Table 1 molecules-27-03829-t001:** Comparison of five light-element imaging modes, including imaging mode, collection angles, and (dis)advantages.

Technique	Imaging Mode	Collection Angles	Advantage	Disadvantaged
LAADF	Convergence	α < θ_inner_ < 50 mrad	Light element imaging Lattice stress imaging	Sample drift and scan distortions
				Affected by thickness
				Accuracy of light elements is insufficient
ABF	Convergence	θ_inner_ = α/2, θ_outer_ = α	High resolution of light elements	Sample drift and scan distortions
				Difficulty in distinguishing element types
				Invalid for weak phases
iDPC	Convergence	4 segment detectors	High resolution of light elements	Sample drift Scan distortions
NCSI	Parallel	CCD	No sample drifts Scan distortions High resolution of light elements	Accuracy of heavy elements is insufficient
				strongly dependent on aberrations and the sample thickness
				Difficulty in interpretation
iSTEM	Convergence	CCD	High resolution of light and heavy elements	Strongly dependent on aberrations and the sample thickness
				Difficulty in interpretation

**Table 2 molecules-27-03829-t002:** Studies on MAPbI3 electron diffraction or fast Fourier transform patterns in the literature.

Specimen	Focus/Direction	Observation Method	Ref.
MAPbI_3_ and MAPbBr_3_	Periodic structure evolution	SAED	[120]
MAPbI_3_	In situ experiment (molecular-level)	SAED	[121]
MAPbI_3_ and MAPb_0.9_Cd_0.1_I_3_	Periodic structure evolution	HRTEM and SAED	[122]
MAPbCl_3_	Periodic structure determines	HRTEM with FFT	[123]
MABr: SnBr_2_: CuCl_2_1:1:0/1:0.5:0.5/1:0:1	Morphology and crystallinity	HRTEM and SAED	[124]
CsPbI_3_ and MAPbI_3_	Morphological, structural and irradiation damage	HRTEM and SAED	[125]
		with accelerated voltage (200 kV/60 kV);	
		dose rate (C1C2) changes	
MAPbBr_3_ quantum dots	Morphology and crystallinity	HRTEM and SAED	[126]
MAPbI_3_	Morphology and structure evolution	HAADF-STEM and SAED	[127]
BA2PbBr4	Periodic structure evolution	SAED	[128]
MAPb_1−x_Pd_x_I_3_ x = 0.25, 0.5	Component distribution	EELS	[129]
MAPbI_3_	Periodic structure evolution	Low-dose HRTEM and SAED	[130]
MAPbI_3_	Photoinduced Degradation	Low-dose HRTEM, SAED	[131]
		HAADF-STEM and EELS	
MAPbI_3_ and MASnI_3_	Periodic structure determines	HRTEM and SAED	[132]
FAPbBr_3_-WS_2_ and MoS_2_	Periodic structure evolution	HRTEM and SAED	[133]
(CsPbI_3_)_0.05_((FAPbI_3_)_1−x_(MAPbBr_3_)_x_)_0.95_	Interface and element distribution	HRTEM and EDS	[134]
BA_2_FAPb_2_I_7_BA_2_MAPb_2_I_7_	Structure evolution	HRTEM and SAED	[135]
		TVIPS camera,	
		Gatan 914.6 cooling holder	
MAPbI_3_	Surface structure and defects	HRTEM and FFT	[136]
		Cryo-TEM	
MAPbI_3_	In situ experiment	Liquid-cell TEM	[137]
MAPbBr_3_	Atomic structure characterization	iDPC-STEM	[138]
MAPbBr_3_	Atomic structure characterization	High-SNR HRTEM image (Dose fractionation)	[139]
		direct electron detector	
(2T)_2_PbI_4_-(2T)_2_PbBr_4_	Surface structure and defects	High-SNR HRTEM image (Dose fractionation)	[140]

## Data Availability

Not applicable.

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
