# Peer review of "The Development of iDPC-STEM and Its Application in Electron Beam Sensitive Materials"

_molecules, 2022, doi:10.3390/molecules27123829_

Round 1
Reviewer 1 Report
The review paper is well-organised and the developments and applications of iDPC-STEM in electron beam sensitive materials are provided. An appropriate outlook on iDPC-STEM capabilities and development is listed. some minor parts that need to be corrected before acceptance:
Figure 7: it is not clear enough to show the word in it.
Line 181-182: Specify what the requirements are.
Figure 12c-12f: The figure mentioned in the paragraph from line 259-273 was not found.
Line 225: please term and specify the what the materials you discussing about. ZSM?
Figure 13c: word missing.
Line 277: ‘Now,….become a reality’. Need to reword
Figure 14b: word not shown completely
Line 296: please term and specify the what the materials you are discussing about. SAPO?
Line 311: what types of loading?
Line 320: define MIL-101
Line 331: define DDEC DQE
Line 333-334: how it greatly advanced?
Line 341-343: Please indicate which figure corresponds to which bracket. The top right corner of (c) was cut. Content on the figure was not clear.
Line 344: Poor resolution. No scale bar.
Line 389: The coordinate axis was cut. No scale bar.
Figure 19: increase loading mass?
Author Response
Response to reviewers
Comments 1: Figure 7: it is not clear enough to show the word in it.
Reply 1: We have improved the image quality.
Comments 2: Line 181-182: Specify what the requirements are.
Reply 2: We have expanded the description in detail, as follows:
In terms of data, iDPC technique does not have a huge dataset. In terms of equipment, iDPC technique only demands an additional four-section detector. Compared with K2 technique, which demands expensive cameras, iDPC technique has simpler equipment requirements and higher cost performance.
Comments 3: Figure 12c-12f: The figure mentioned in the paragraph from line 259-273 was not found.
Reply 3:We have simplified the figures and changed the annotation, as follows
- a) iDPC-STEM image of Mo/ZSM-5; b) Zoomed-in areas 1(â… ), 2(â…¡), and 3(â…¢) of (a) with an atomic model
Comments 4: Line 225: please term and specify the what the materials you discussing about. ZSM?
Reply 4: We have a more precise definition of ZSM-5 material, as follows:
ZSM-5 is a medium pore zeolite containing ten-membered rings. The basic structural unit is composed of eight five-membered rings. The framework is composed of two intersecting channel systems. The short axis is 5.1-5.2 Å; the other is a "Z"-shaped transverse channel with a nearly circular cross-section and a pore diameter of 5.4 ± 0.2 Å. Its unique microporous structure makes ZSM-5 have excellent shape selectivity and is a commonly used catalyst in the chemical field. ZSM-5 can be used in a variety of aromatization reactions, such as the methanol-to-aromatics process (MTA).
Comments 5: Figure 13c: word missing.
Reply 5: We changed the figure.
Comments 6:Line 277: ‘Now,….become a reality’.Need to reword
Reply 6: We have deleted this sentence and expressed its meaning by modifying the description of the iDPC-STEM technique in the full text.
Comments 7:Figure 14b: word not shown completely
Reply 7: We changed the figure.
Comments 8:Line 296: please term and specify the what the materials you are discussing about. SAPO?
Reply 8: We describe it more accurately
Silicoaluminophosphate (SAPO) molecular sieves with small pores, such as SAPO-34 and SAPO-18, play an important catalytic role in methanol-to-olefin (MTO) reactions. Among them, molecular sieves with SAPO-18/SAPO-34 eutectic structure often show better performance than single molecular sieves in catalytic reactions due to the presence of both AEI and CHA structural units. However, due to the difficulty of electron microscopy characterization, SAPO-18 The /SAPO-34 eutectic structure is less studied.
Comments 9:Line 311: what types of loading?
Reply 9: We've changed the way we describe it, removing the description of the loading, as follow:
In the field of zeolite, iDPC-STEM has achieved the characterization of fine structure. While providing real-space evidence that was difficult to obtain in the past, it has carried out a more in-depth analysis of the relationship between structure and catalytic performance, which has a relatively complete and systematic study and has great development potential.
Comments 10:Line 320: define MIL-101
Reply 10: Defined as follows
MIL-101, Cr3F(H2O)O[(O2C)C6H4(CO2)]3·nH2O(n≈25), the specific surface area is mostly above 4000 m2/g, and there is an effective inner diameter of about 2.9 nm and a window diameter of about 2.9 nm. Pentagonal holes with a diameter of about 1.2 nm and hexagonal holes with an effective inner diameter of about 3.4 nm and a window diameter of about 1.4 nm. These structural features endow MIL101 with unique advantages, making it one of the most potential MOFs materials. MIL-101 is commonly used to support various metal atoms or active particles in catalysis research. Therefore, if the local structure of MIL-101 (the coordination relationship of nodes and linkers, etc.) can be imaged directly at the atomic scale, it would help understand the relationship between catalyst structure and performance.
Comments 11:Line 331: define DDEC DQE
Reply 11:Defined as follows
direct detection electron counting (DDEC)
detective quantum efficiency (DQE)
Comments 12:Line 333-334: how it greatly advanced?
Reply 12:We delete this sentence and show the impact of iDPC-STEM on imaging quality through the analysis of application.
Comments 13:Line 341-343: Please indicate which figure corresponds to which bracket. The top right corner of (c) was cut. Content on the figure was not clear.
Reply 13: We changed the typography and now it is visible.
Comments 14:Line 344: Poor resolution. No scale bar.
Reply 14: The MOF is susceptible to electron beam damage, and the electron beam is confined to protect the sample, resulting in reduced resolution. We changed the figure.
Comments 15:Line 389: The coordinate axis was cut. No scale bar.
Reply 15: We changed the figures.
Comments 16:Figure 19: increase loading mass?
Reply 16: We changed the figures.

Reviewer 2 Report
Review paper entitled : Development of iDPC-STEM and its Application in Electron Beam Sensitive Materials
In this work, a review of the developments and applications of iDPC-STEM in electron beam sensitive materials, and an outlook on its capabilities and development is presented.
Several examples of application have been presented. On the other hand, it is useful to explain in more detail the limits of this technique.
Q1 The quality of the figure 8 must be improved.
Q2 In recent years, with the gradual popularization of iDPC-STEM, it has played an increasingly important role in the characterization of the surface, interface, and defects different materials. do you have a map concerning the existence of these microsppes in the world?
Q3 What about the required sample preparation for iDPC-STEM?
Q4 The majority of the data presented in this work is focused on images constrast and quality. No eels, no microanalysis, ... are presented.
Q5 It is announced that iDPC-STEM signals are the consequence of interactions between a focused electron beam and the electric field of a sample. No things are announced about the magnetic moment contribution.
Author Response
Response to reviewers
Comments 1:The quality of the figure 8 must be improved.
Reply 1: We have optimized the image.
Comments 2: In recent years, with the gradual popularization of iDPC-STEM, it has played an increasingly important role in the characterization of the surface, interface, and defects different materials. do you have a map concerning the existence of these microsppes in the world?
Reply 2: Thank you for your suggestion, the amount of this equipment in reality has a very important impact. After research, iDPC-STEM only requires an additional detector, in the new transmission electron microscope equipment, usually have this function. It is also one of the advantages of this technique: low requirements for equipment. Therefore, it is difficult to accurately assess the distribution of this equipment in the world. Its application is limited mainly because the technology has just emerged, and the scope of its ability and operation methods are rarely understood. It is the purpose of this paper to solve this problem.
Comments 3:What about the required sample preparation for iDPC-STEM?
Reply 3: The preparation of iDPC-STEM sample is the same as that of traditional electron microscope sample. In the field of biomaterials, the high contrast of iDPC-STEM can be used to carry out structural characterization with little or no staining, it is probably the biggest difference in sample preparation.
Comments 4:The majority of the data presented in this work is focused on images constrast and quality. No eels, no microanalysis, ... are presented.
Reply 4: Thank you for your suggestion. As you said, the structural characterization of materials cannot be limited to the analysis of image contrast, and more and more comprehensive relevant information is needed. Therefore, iDPC-STEM technology must be combined with more analysis methods in the future. However, for iDPC-STEM, an imaging technology, all we can obtain is the relevant information extended by the image contrast. (Though the relationship between the image contrast and the atomic number can be used to conduct a rough qualitative analysis of the composition) The relevant work on the more detailed analysis of the microscopic components by EELS can be summarized in the next article.
Comments 5:It is announced that iDPC-STEM signals are the consequence of interactions between a focused electron beam and the electric field of a sample. No things are announced about the magnetic moment contribution.
Reply 5: Thank you for your suggestion. According to the imaging principle of iDPC-STEM, the magnetic moment must have an influence on its imaging, which has been verified when we conduct iDPC-STEM experiments: Using samples with magnetic (a little) will make the obtained iDPC-STEM images difficult to reflect the atomic structure, while HAADF-STEM does. Since the technique itself is relatively new and this direction is still developing, the relevant data we can summarize is limited, we will invest more energy in this field in the future.
